# Frequent Onsets of Cellulitis in Lower Limbs with Lymphedema Following COVID-19 mRNA Vaccination

**DOI:** 10.3390/vaccines10040517

**Published:** 2022-03-26

**Authors:** Tatsuma Okazaki, Momoko Matashiro, Gaku Kodama, Takeshi Tshubota, Yoshihito Furusawa, Shin-Ichi Izumi

**Affiliations:** 1Department of Physical Medicine and Rehabilitation, Graduate School of Medicine, Tohoku University, Sendai 980-8575, Japan; yoshihito.furusawa.b2@tohoku.ac.jp (Y.F.); izumis@med.tohoku.ac.jp (S.-I.I.); 2Department of Rehabilitation, Tohoku University Hospital, Sendai 980-8574, Japan; momo0605lami@yahoo.co.jp (M.M.); gaku_kodama@yahoo.co.jp (G.K.); takeshi.tsubota.b1@tohoku.ac.jp (T.T.); 3Center for Dysphagia of Tohoku University Hospital, Sendai 980-8575, Japan; 4Department of Physical Medicine and Rehabilitation, Graduate School of Biomedical Engineering, Tohoku University, Sendai 980-8575, Japan

**Keywords:** cellulitis, lymphedema, COVID-19 mRNA vaccination

## Abstract

Four patients with secondary lower limb lymphedema developed cellulitis at their lymphedema lesion following COVID-19 mRNA vaccinations. They did not develop adverse effects at their vaccination site. All the patients were Japanese females aged <60 years. Three patients developed cellulitis following the first vaccination. The date of onset of cellulitis following the first vaccination varied from 0 to 21 days. Two received BNT162b2 mRNA vaccines and the others received mRNA-1273 vaccines. All the patients were treated with oral antibiotics and recovered. Two patients had repeated cellulitis. The patients with the repeated development of cellulitis could not perform good skincare. One patient had joint contractures in their lower limbs and could not reach her lymphedema lesions, and the other patient could not master the skincare. According to previous studies, the development of cellulitis following vaccination was rare. In this study, four patients aged <60 years developed cellulitis among the eight patients that regularly visited our hospital for rehabilitation for their lower limb lymphedema. In patients with lymphedema, prolonged inflammation may impair lymphatic functions and worsen edema. Therefore, at the time of vaccination, we should keep in mind the prevention and immediate management of cellulitis using intensive skincare and antibiotic treatment.

## 1. Introduction

Randomized clinical trials suggested that a risk factor for developing adverse effects following COVID-19 mRNA vaccination was younger age [1,2,3]. Additionally, a recent online cohort study in the United States suggested Asian ancestry and female sex as risk factors [1]. Although further worldwide studies and analysis are necessary to identify the risk factors, Japanese females aged <60 years with lymphedema developed cellulitis following COVID-19 mRNA vaccinations. There are a few reports of infection as an adverse effect following mRNA-1273 (Moderna Inc., Cambridge, MA, USA) and BNT162b2 (Pfizer Inc., New York, NY, USA and BioNTech, Mainz, Germany) vaccinations, including cellulitis. In Japan, there were 3 registered cases of cellulitis following 32,433,544 mRNA-1273 vaccinations and 35 cases following 168,296,343 BNT162b2 mRNA vaccinations. Therefore, the frequency was less than 1 in a million vaccinations. In this regard, the site of cellulitis was not reported, and a case report showed the onset of cellulitis at an injection site [4].

Eight patients were regularly visiting our hospital for rehabilitation of lymphedema of the lower limbs. All the patients were Japanese females aged <60 years. Among them, four patients developed cellulitis at their lymphedema lesions following mRNA-1273 or BNT162b2 mRNA vaccinations. This report aims to highlight the frequent onset of cellulitis following COVID-19 mRNA vaccination in patients with lymphedema.

## 2. Cases

The first case was a 52-year-old female suffering from a lower left limb lymphedema for 12 years. She took care of her skin very carefully and had no history of cellulitis. She underwent stage 1b endometrial cancer surgery and developed lymphedema 5 years after the surgery. She underwent lymphaticovenular anastomosis surgery 3, 4, and 5 years ago. She developed cellulitis 11 days after the first BNT162b2 mRNA vaccination (Figure 1A) and took 750 mg/day of the oral antibiotic cefaclor for 5 days to recover from the cellulitis. Three days after the onset, a laboratory study showed an elevated C-reactive protein level (CRP) 2.4 mg/dL (normal rage < 0.14 mg/dL) and a leukocyte count within the normal range, 4960/μL (normal range 3300–8600/μL). The onset of the cellulitis did not repeat and she received the second vaccination.

The second case was a 52-year-old female with lymphedema of both lower limbs. She has been suffering from lymphedema for seven years as an adverse effect of docetaxel for managing stage 4 breast cancer. She developed her first cellulitis three years ago. Before the vaccination, she developed cellulitis five times in two and a half years. Following the mRNA-1273 vaccination, she developed cellulitis four times in five months. She developed cellulitis once following the first vaccination and three times following the second vaccination. On the 21st day following the first vaccination, she developed cellulitis and recovered by taking an oral antibiotic, cefaclor (750 mg/day), for 7 days. In the 2nd, 3rd, and 4th months following the 2nd vaccination, she developed cellulitis. Laboratory data of her 2nd cellulitis following the vaccination, which developed in the 2nd month following the 2nd vaccination, showed an elevated CRP level 9.49 mg/dL (normal rage < 0.14 mg/dL) and a leukocyte count within the normal range, 5700/μL (normal range 3300–8600/μL). Her eosinophil number was within the normal range at 140/μL (normal range 30–600/μL). She took 750 mg/day of the oral antibiotic cefaclor for 7 days and recovered. She developed her 5th cellulitis in the 4th month following the 2nd vaccination, on the admission day for intensive lymphedema management. Her entire right lower limb showed mild redness and she felt sick. Laboratory data showed an elevated CRP level at 1.02 mg/dL (normal rage < 0.14 mg/dL), a normal range leukocyte count of 5700/μL (normal range 3300–8600/μL), an eosinophil count of 170/μL (normal range 30–600/μL), and a normal body temperature. D-dimer was within normal range at <0.5 μg/dL, suggesting a low possibility of thrombosis. She took amoxicillin (2000 mg/day) and clavulanic acid (500 mg/day) for 7 days and recovered from cellulitis with a CRP level of 0.21 mg/dL (normal rage < 0.14 mg/dL).

The third case was a 49-year-old female who underwent stage 2A cervical cancer surgery 9 years ago and suffered from a lower left limb lymphedema for 8 years. She underwent lymphaticovenular anastomosis surgery 1, 4, and 5 years ago and underwent lung surgery for lung tumors 4 and 5 years ago. Before the vaccination, she developed cellulitis four times; 1, 2, 4, and 6 years ago. She developed cellulitis the same day following the second BNT162b2 vaccination (Figure 1B,C) and took minocycline (200 mg/day) for 7 days to recover from the cellulitis.

The fourth case was a 45-year-old female who underwent surgery for stage 1A cervical cancer and was suffering from lymphedema for 4 years. She received no particular treatment for the edema. However, she began to develop cellulitis following the vaccination and visited the lymphedema clinic. Table 1 shows the overall characteristics of the above patients.

## 3. Discussion

Four patients with secondary lymphedema on lower limbs developed cellulitis following the COVID-19 mRNA vaccination. Cellulitis did not develop at their injection site but the lymphedema site. Currently, eight lower limb lymphedema patients aged <60 years regularly visited our hospital for rehabilitation. Therefore, cellulitis developed among 50% of these patients, whereas among registered adverse effects in Japan, the frequency of cellulitis was less than one following a million vaccinations. Two of the patients developed cellulitis for the first time. Moreover, two patients repeated the development of cellulitis after the vaccination. Three patients developed cellulitis after the first vaccination and one after the second.

After the onset of cellulitis, three out of four patients became more cautious of their skincare than before. However, one had joint contracture in the lower limbs and, as a result, she could not reach the peripheral side of her lower limbs nor perform good skincare and the cellulitis development repeated. One patient could not master the skincare and also experienced repeated development.

According to previous studies, the development of cellulitis following the vaccination was rare [5,6]. Among healthcare professionals, BNT162b2 mRNA vaccination developed cellulitis in 0 out of 1245 subjects [7] and the mRNA-1273 vaccination developed cellulitis in 0 out of 1116 subjects [8]. Some reports showed the development of cellulitis at the local vaccination site but not at the other site [4,9]. We had CRP data for only two out of four patients, but the CRP levels were both elevated and the numbers of eosinophils were within the normal range. A study with cutaneous allergic reactions after the vaccinations reported that 63% of the cases developed the reactions only after the second vaccination but did not at the first vaccination [10]. In contrast, three out of four patients developed skin reactions after the first vaccination in this study. Therefore, we interpreted that the reaction was not allergic as those sensitized at the first vaccination developed an allergic reaction at the second. Moreover, the effectiveness of treatment with oral antibiotics alone strengthened the diagnosis of cellulitis rather than an allergic reaction. In this study, patients have not shown any history of autoimmune diseases or laboratory study data of chronic inflammation. Since we were afraid of performing excessive examinations, we could not further evaluate autoimmune markers for these patients. This concern also applied to a patient with a thrombosis diagnosis and a normal range D-dimer. Therefore, we could not further examine her lower extremities, such as with an ultrasound.

To identify the mechanism of frequent onset of cellulitis at lymphedema lesions, we checked the adverse effects of COVID-19 mRNA vaccinations and would suggest hyaluronan accumulation as one of the potential causes. Hyaluronan may accumulate at the lymphedema lesions and may amplify inflammation. Hyaluronan is often used as a dermal filler and inflammatory reactions to hyaluronan and dermal fillers have been reported as an adverse effect of the COVID-19 mRNA vaccine [10,11]. In addition, lymphatics are the central route for hyaluronan drainage [12,13]. Indeed, hyaluronan accumulated at the tissues with dysfunctional lymphatics, such as lymphedema and tumors [12,13]. Overall, following COVID-19 mRNA vaccination, the tissues with hyaluronan accumulation might become highly reactive to immunogens and develop cellulitis in response to the invasion of low immunogenic pathogens that hosts normally do not react to.

Initially, we speculated that adjuvants in the vaccines accumulated at the lymphedema site due to their impaired drainage function and caused susceptibility to cellulitis. However, the mRNA vaccines do not contain adjuvants.

Patients did not show apparent psychological impact associated with cellulitis following the vaccination, such as regretting or evasion of vaccinations. 

A previous study reported that a man developed minimal change disease with nephrotic syndrome, acute kidney injury, and anasarca following the BNT162b2 mRNA vaccination [14]. In our study, three patients were examined for serum creatinine levels after the development of cellulitis and all were within the normal range (normal range 0.46–0.79 mg/dL). However, the examination dates after the cellulitis varied from the day of to 3 months after the onset. Two patients underwent urinalysis and showed protein (-) and blood (-). The examined dates were the day of and 3 months after the onset. No patients showed new peripheral edema, anasarca, or dyspnea. These data suggest a low possibility of developing renal dysfunction following the vaccine in patients in this study. 

Previous studies suggested the mechanisms of adverse reactions following COVID-19 mRNA vaccinations as delayed-type or T-cell-mediated hypersensitivity [9]. Another study suggested early T cell-mediated injury in response to the vaccine [14]. In our study, oral antibiotics were effective for all the patients. We may have to consider further examinations, such as a skin biopsy, when we experience patients with inflammation in their lymphedema lesions refractory to antibiotics in the future. 

Inflammations induce abnormal morphology of lymphatics, impair their drainage function, and worsen lymphedema [15]. In addition, lymphatics with abnormal morphology impaired immunological function in cancer models [13]. Therefore, they might also impair immunological function in infectious diseases such as cellulitis. Thus, we should immediately manage cellulitis at lymphedema lesions.

## 4. Conclusions

Patients with lower limb lymphedema received COVID-19 mRNA vaccinations. They were Japanese females aged <60 years and visiting a hospital regularly for lymphedema rehabilitation. Four patients developed cellulitis at the site of the lower limb lymphedema. All four patients suffered from secondary lymphedema. Oral antibiotics were effective for all the patients. However, prolonged inflammation may impair lymphatic functions and worsen edema in patients with lymphedema. Therefore, at the time of the vaccination, healthcare workers and patients should keep in mind the prevention and immediate management of cellulitis by intensive skincare and antibiotic treatment.

## Figures and Tables

**Figure 1 vaccines-10-00517-f001:**
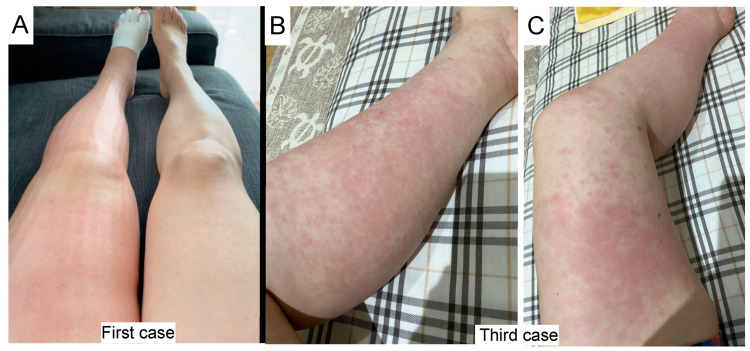
(**A**) The first case. A 52-year-old female with lymphedema of the lower left limb developed cellulitis at her lymphedema lesion 11 days after the first BNT162b2 mRNA vaccination. Mild redness in the lower left leg. Compression therapy by an elastic garment made wrinkles on the left leg, and an elastic bandage wrapped the left toe. (**B**,**C**) The third case. A 49-year-old female with lymphedema of the lower left limb developed cellulitis at her lymphedema lesion on the day of the second BNT162b2 vaccination. (**B**) Development of cellulitis on the lower leg. (**C**) Development of cellulitis on the entire lower limb.

**Table 1 vaccines-10-00517-t001:** Overall characteristics of the patients.

	Age	Vaccination Type	History of Cellulitis Before the Vaccination	Repetition of Cellulitis	Onset after First or Second Vaccination
Case 1	52	BNT162b2 mRNA	No	No	First vaccination
Case 2	52	mRNA-1273	Yes	Yes	First vaccination
Case 3	49	BNT162b2 mRNA	Yes	No	Second vaccination
Case 4	45	mRNA-1273	No	Yes	First vaccination

## Data Availability

The data in this study are available from the corresponding author upon reasonable request.

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
