# Peer review of "Frequent Onsets of Cellulitis in Lower Limbs with Lymphedema Following COVID-19 mRNA Vaccination"

_vaccines, 2022, doi:10.3390/vaccines10040517_

Round 1

Reviewer 1 Report

The submission “Frequent onsets of cellulitis at lower limbs with lymphedema following COVID-19mRNA vaccination” by Okazaki et al., seem interesting and new at least in Japan. Associated the cellulitis with lymphedema, as a most common complication, is not only reported post COVID-19 vaccinations but it also reported during COVID-19 infection. Often, lymphedema affects people with cancer, as a lymph node dissection following cancer resection. The Okazaki et al report repeating again the alarm of COVID-19 vaccine safety should be considered as long as the vaccine efficacy.

It is recommended to publish this case report in vaccines to increase the scientific knowledge and the decision-maker awareness on the safety of the vaccine at least in combordites patients.

There is a number of major comments that should be fixed before acceptance.

major comments that should be fixed before acceptance.

  1. Those patients’ lymphedema belongs to primary or secondary.
  2. Is there any psychological impact associated with those patients?
  3. Do you evaluate any autoimmune markers for your patients? As all your patients are females.
  4. Please add “oral antibiotics” …. overall manuscript.
  5. Page 1, lines 30-31, Risk factors for the development of adverse effects following mRNA Covid-19 vaccination were Asians, females, and young people [1]. Although the cited reference, there is no consensus considered “Asians, females, and young people” as a risk factors. The issue needs a furthermore studies and analysis.
  6. Case 2 cellulitis is the most complicated one post-vaccination. However, the author's case presentation is overlapping page 2, lines 63-66 “Following the first mRNA- 62 1273 vaccination, she developed cellulitis 4 times in 5 months. On the 21st day following the first vaccination, she developed cellulitis and recovered by taking antibiotics, cefaclor 750 mg/day for 7 days. Following the 2nd vaccination, she developed cellulitis 2, 3, and 4 months later”, please be clearest and detailed during the first/second vaccine dose over time description.
  7. Page 2, line 74“suggesting a low possibility of thrombosis” do you scan or used a lab test?
  8. Please add the normal values for all markers you assessed, such as (eosinophil normal value 30–600/uL).
  9. The elevated eosinophil in case 2 will be categorized as eosinophil cellulitis? Please describe this even in your discussion.

Author Response

  1. Those patients’ lymphedema belongs to primary or secondary.

Response to the reviewer’s comment: Thank you for pointing out this critical issue.  As you commented, we did not refer to primary or secondary.  All the patients suffered from secondary lymphatic edema as we had written their primary diseases in the result. We add “secondary” to the manuscript as shown below;

Page 1, line 13; Four patients with secondary lower limb lymphedema

Page 3, line 104; Four patients with secondary lymphedema

Page 5, line 178; All 4 patients suffered from secondary lymphedema.

  1. Is there any psychological impact associated with those patients?

Response to the reviewer’s comment: Thank you for your in-depth and creative comment.  There was no psychological impact associated with those patients.  The patients did not regret receiving the vaccinations.  All the patients did not reject second vaccinations.  We described these issues as shown below.

Page 4, line 151-152; Patients did not show apparent psychological impact associated with cellulitis following the vaccination, such as regretting or evasion of vaccinations.

  1. Do you evaluate any autoimmune markers for your patients? As all your patients are females.

Response to the reviewer’s comment: We are very sorry for providing insufficient information about the patients.  The oral antibiotics were effective for all the patients.  Moreover, patients have not shown any history of autoimmune diseases or laboratory study data of chronic inflammation.  Therefore, we did not further evaluate autoimmune markers.  We were afraid of being pointed out for excessive examinations.  We described these issues as shown below.

Page 4, line 130-133; In this study, patients have not shown any history of autoimmune diseases or laboratory study data of chronic inflammation. Since we were afraid of being pointed out for excessive examinations, we could not further evaluate autoimmune markers for these patients.

  1. Please add “oral antibiotics” …. overall manuscript.

Response to the reviewer’s comment: Thank you for adding clarity to our manuscript.  We added “oral antibiotics” overall the manuscript. 

  1. Page 1, lines 30-31, Risk factors for the development of adverse effects following mRNA Covid-19 vaccination were Asians, ... Although the cited reference, there is no consensus considered “Asians, females, and young people” as a risk factors. The issue needs a furthermore studies and analysis.

Response to the reviewer’s comment: We are very sorry for our insufficient writing of the previous studies.  We completely agree with your comment.  The majority of participants in the cited papers were Americans and Japanese were the minority.  We wrote these issues as shown below.

Page 1, line 31-36; Randomized clinical trials suggested that a risk factor for developing adverse effects following COVID-19 mRNA vaccination was younger age [1-3]. Additionally, a recent online cohort study in the United States suggested Asians and females as risk factors [1]. Although further worldwide studies and analysis are necessary to identify the risk factors, Japanese females aged <60 years with lymphedema developed cellulitis following COVID-19 mRNA vaccinations.

  1. Case 2 cellulitis is the most complicated one post-vaccination. However, the author's case presentation is overlapping page 2, lines 63-66 “Following the first mRNA- 62 1273 vaccination, she developed cellulitis 4 times in 5 months. On the 21st day following the first vaccination, she developed cellulitis and recovered by taking antibiotics, cefaclor 750 mg/day for 7 days. Following the 2nd vaccination, she developed cellulitis 2, 3, and 4 months later”, please be clearest and detailed during the first/second vaccine dose over time description.

Response to the reviewer’s comment: We are very sorry for confusing you with first/second vaccine dose over time description.  We fixed the writing to be clearest and detailed, as shown below.  We believe this fixation helps you and readers to understand the manuscript without difficulties.

Page 2, line 72-Page 3, line 83; She developed cellulitis once following the first vaccination and 3 times following the second vaccination. On the 21st day following the first vaccination, she developed cellulitis and recovered by taking oral antibiotics, cefaclor 750 mg/day, for 7 days. In the 2nd, 3rd, and 4th months following the 2nd vaccination, she developed cellulitis. Laboratory data of her 2nd cellulitis following the vaccination, which was developed in the 2nd month following the 2nd vaccination, showed elevated CRP level 9.49 mg/dL (normal rage <0.14 mg/dL) and leukocyte count within the normal range 5700/μL (normal range; 3300-8600/μL). An eosinophil number was within the normal range 140/μL (normal range 30-600/μL). She took 750 mg/day of the oral antibiotic cefaclor for 7 days and recovered. She developed her 5th cellulitis following the vaccination, which was in the 4th month following the 2nd vaccination,

  1. Page 2, line 74“suggesting a low possibility of thrombosis” do you scan or used a lab test?

Response to the reviewer’s comment: Thank you for pointing out this issue.  The D-dimer was within a normal range.  Since we were afraid of being pointed out for excessive examinations, we could not evaluate further data.  We wrote this issue as shown below.

Page 4, line 133-136; Since we were afraid of being pointed out for excessive examinations, we could not further evaluate autoimmune markers for these patients. This concern was also applied for a thrombosis diagnosis in a patient with a normal range D-dimer. Therefore, we could not further examine her lower extremities, such as by an ultrasound.

  1. Please add the normal values for all markers you assessed, such as (eosinophil normal value 30–600/uL).

Response to the reviewer’s comment: Thank you for pointing out this issue.  We added the normal values for all markers we assessed.

 The elevated eosinophil in case 2 will be categorized as eosinophil cellulitis? Please describe this even in your discussion.

Response to the reviewer’s comment: We are very sorry for our confusing writing.  The number of eosinophils was within normal range.  To avoid misunderstanding, we fixed the description in the result, as shown below.  We also add a description in the discussion.

Page 3, line 84-87; Laboratory data showed elevated CRP level 1.02 mg/dL (normal rage <0.14 mg/dL), and within normal range data leukocyte count 5700/μL (normal range; 3300-8600/μL) with eosinophil count 170/μL (normal range 30-600/μL) and normal body temperature.

Page 4, line 167-169; We may have to consider further examinations such as a skin biopsy when we experience patients with inflammation in their lymphedema lesions refractory to antibiotics in the future. 

Reviewer 2 Report

Okazaki et al. reported that “Frequent onsets of cellulitis at lower limbs with lymphedema fol-2 lowing COVID-19mRNA vaccination”.

  1. Authors used both COVID-19 and Covid-19. Please use COVID-19.
  2. “…In our study, 4 out of 8 regular hospital visiting patients…” What is 8?
  3. Authors should write company name, city, country after BNT162b2 mRNA and mRNA-1273.
  4. Combine Fig 1 and Fig 2 into one figure.
  5. Please make one table demonstrating laboratory data of 4 patients.
  6. Lebedev et al. reported that minimal change disease following the Pfizer-BioNTech COVID-19 vaccine. How the renal function of these four patients and urinary findings? Authors should discuss about them more.
  7. How the mechanism of the cellulitis developing?? Authors should discuss more.

Author Response

  1. Authors used both COVID-19 and Covid-19. Please use COVID-19.

Response to the reviewer’s comment: We are very sorry for the mixed-use of COVID-19 and Covid-19.  We fixed all the Covid-19 to COVID-19.

  1. “…In our study, 4 out of 8 regular hospital visiting patients…” What is 8?

Response to the reviewer’s comment: We are very sorry for the vague description.  We fixed the description as shown below.

Page 1, line 22-24; In this study, 4 patients developed cellulitis among 8 patients that regularly visited our hospital for rehabilitation for their lower limb lymphedema aged <60 years.

  1. Authors should write company name, city, country after BNT162b2 mRNA and mRNA-1273.

Response to the reviewer’s comment: We are very sorry for lacking a detailed description of the mRNA vaccines.  We wrote company name, city, state, or country as shown below.

Page 1, line 36-38; As an adverse effect following mRNA-1273 (Moderna Inc., Cambridge, MA) and BNT162b2 (Pfizer Inc., New York, NY and BioNTech, Mainz, Germany) vaccinations, there are a few reports of infection, including cellulitis.

  1. Combine Fig 1 and Fig 2 into one figure.

Response to the reviewer’s comment: Thank you for your suggestion.  We combined Fig.1 and Fig. 2.  They are shown as Fig1A-C.

  1. Please make one table demonstrating laboratory data of 4 patients.

Response to the reviewer’s comment: Thank you for your creative comment.  The patients visited our hospital for rehabilitation of the lymphedema and did not always visit us for cellulitis.  Sometimes we treated their cellulitis.  However, they visited nearby clinics for cellulitis also.  Laboratory exams were not always performed.  Since patients have official medicine notebooks approved by the Japanese health insurance system, we were able to confirm the disease name and the medicine in addition to the patients’ declaration.  Thus, we do not have all the cellulitis data and could not make a table you suggested.  We are very sorry that we cannot respond to your suggestion.

  1. Lebedev et al. reported that minimal change disease following the Pfizer-BioNTech COVID-19 vaccine. How the renal function of these four patients and urinary findings? Authors should discuss about them more.

Response to the reviewer’s comment: Thank you for pointing out this issue.  Three patients had serum creatinine data, and 2 patients had urinalysis data.  All the data were within normal range.  In addition, they did not show symptoms of renal dysfunction such as extra edema or dyspnea.  We discussed these issues in discussion as shown below.

Page 4, line 153-161; A previous study reported that a man developed minimal change disease with nephrotic syndrome, acute kidney injury, and anasarca following BNT162b2 mRNA vaccination [14]. In our study, 3 patients were examined for serum creatinine levels after the development of the cellulitis, which all were within the normal range (normal range; 0.46-0.79 mg/dL). However, the examination dates after the cellulitis varied from the day of the onset to 3 months after the onset. Two patients underwent urinalysis and showed protein (-) and blood (-). The examined dates were the day of and 3 months after the onset. No patients showed new peripheral edema, anasarca, or dyspnea.  These data suggest a low possibility of developing renal dysfunction following the vaccinations in patients in this study.

  1. How the mechanism of the cellulitis developing?? Authors should discuss more.

Response to the reviewer’s comment: Thank you for your in-depth comment.  We discussed more in the discussion, as shown below.

Page 4, line 162-168; Previous studies suggested mechanisms of adverse reactions following COVID-19 mRNA vaccinations as delayed-type or T-cell-mediated hypersensitivity [9]. Another study suggested early T cell-mediated injury in response to the vaccine [14]. In our study, oral antibiotics were effective for all the patients. We may have to consider further examinations such as a skin biopsy when we experience patients with inflammation in their lymphedema lesions refractory to antibiotics in the future. 

Round 2

Reviewer 1 Report

Thank you very much, authors almost response properly and improve their manuscript. Thank you

Reviewer 2 Report

All queries have been addressed.